# Genome-Wide Identification of Auxin-Responsive *GH3* Gene Family in *Saccharum* and the Expression of *ScGH3-1* in Stress Response

**DOI:** 10.3390/ijms232112750

**Published:** 2022-10-22

**Authors:** Wenhui Zou, Peixia Lin, Zhennan Zhao, Dongjiao Wang, Liqian Qin, Fu Xu, Yachun Su, Qibin Wu, Youxiong Que

**Affiliations:** 1Key Laboratory of Sugarcane Biology and Genetic Breeding, Ministry of Agriculture and Rural Affairs, Fujian Agriculture and Forestry University, Fuzhou 350002, China; 2Key Laboratory of Genetics, Breeding and Multiple Utilization of Crops, Ministry of Education, College of Agriculture, Fujian Agriculture and Forestry University, Fuzhou 350002, China

**Keywords:** sugarcane, GH3 gene family, genome-wide analysis, RT-qPCR, disease resistance

## Abstract

Gretchen Hagen3 (*GH3*), one of the three major auxin-responsive gene families, is involved in hormone homeostasis in vivo by amino acid splicing with the free forms of salicylic acid (SA), jasmonic acid (JA) or indole-3-acetic acid (IAA). Until now, the functions of sugarcane *GH3* (*SsGH3*) family genes in response to biotic stresses have been largely unknown. In this study, we performed a systematic identification of the *SsGH3* gene family at the genome level and identified 41 members on 19 chromosomes in the wild sugarcane species, *Saccharum spontaneum*. Many of these genes were segmentally duplicated and polyploidization was the main contributor to the increased number of *SsGH3* members. SsGH3 proteins can be divided into three major categories (SsGH3-I, SsGH3-II, and SsGH3-III) and most *SsGH3* genes have relatively conserved exon-intron arrangements and motif compositions. Diverse *cis*-elements in the promoters of *SsGH3* genes were predicted to be essential players in regulating *SsGH3* expression patterns. Multiple transcriptome datasets demonstrated that many *SsGH3* genes were responsive to biotic and abiotic stresses and possibly had important functions in the stress response. RNA sequencing and RT-qPCR analysis revealed that *SsGH3* genes were differentially expressed in sugarcane tissues and under *Sporisorium scitamineum* stress. In addition, the *SsGH3* homolog *ScGH3-1* gene (GenBank accession number: OP429459) was cloned from the sugarcane cultivar (*Saccharum* hybrid) ROC22 and verified to encode a nuclear- and membrane-localization protein. *ScGH3-1* was constitutively expressed in all tissues of sugarcane and the highest amount was observed in the stem pith. Interestingly, it was down-regulated after smut pathogen infection but up-regulated after MeJA and SA treatments. Furthermore, transiently overexpressed *Nicotiana benthamiana*, transduced with the *ScGH3-1* gene, showed negative regulation in response to the infection of *Ralstonia solanacearum* and *Fusarium solani* var. *coeruleum*. Finally, a potential model for *ScGH3-1*-mediated regulation of resistance to pathogen infection in transgenic *N. benthamiana* plants was proposed. This study lays the foundation for a comprehensive understanding of the sequence characteristics, structural properties, evolutionary relationships, and expression of the *GH3* gene family and thus provides a potential genetic resource for sugarcane disease-resistance breeding.

## 1. Introduction

Growth hormones are important to plant development since they not only affect cell division and value addition in plants but are also involved in phototropism and gravitropism [1]. The auxin early-response genes can be roughly divided into three categories: Aux/IAAs (auxin/indoleacetic acid), SAURs (small auxin up-RNAs), and GH3s (Gretchen Hagen 3), which are capable of inducing the rapid and high expression of certain genes [2,3]_._ The GH3 protein is involved in hormone homeostasis in vivo by amino acid splicing, with the free forms of salicylic acid (SA), jasmonic acid (JA), or indole-3-acetic acid (IAA) [4]. The *GH3* gene was first discovered in soybean (*Glycine max*) in 1987 [5] and was found later in *Arabidopsis* (*Arabidopsis thaliana*) [6], tobacco (*Nicotiana tabacum*) [7], rice (*Oryza sativa*) [8], wheat (*Triticum aestivum*) [9] and maize (*Zea mays*) [10]. The functional study of the plant *GH3* gene family has become a hot topic.

In plants, *GH3* genes are key components of the hormonal machinery regulating growth and development in response to biotic and abiotic stresses [4]. GH3 is a phytohormone-responsive protein found in many plants; these proteins play important roles in enhancing the biological activities of IAA, JA, and SA [11]. In *Arabidopsis*, the *JAR1* gene not only resisted a soil *Pythium debaryanum* infestation but also reduced ozone damage [12,13,14,15]. The *AtGH3a/GH3.5* gene regulated the expression of gene *PR-1*, which was involved in the regulation of the SA-mediated plant defense response [16]. Similarly, the *PBS3/GH3.12* knockout *Arabidopsis* showed more pronounced disease than the wild type when infested with the *Pseudomonas syringae* pathogen DC3000 (*avrPphB*) [17], while the concentrations of active glucoside-conjugated SA (SAG) were reduced in the mutants [17]. In rice, the *OsGH3.8* gene could activate resistance to *Xanthomonas oryzae* pv. *oryzae* (*Xoo*) independently of the SA and JA signaling pathways; the overexpression of this gene not only increased resistance to the pathogen but also suppressed the expression of expansion, while this inhibition was not only related to the control of disease infestation but also led to morphological changes and the delayed growth and development of rice [18]. Hui et al. [19] catalyzed the conversion of free JA to the active form of JA-Ile and verified that four members of the group I *GH3* family, namely, *GH3.3*, *GH3.5*, *GH3.6*, and *GH3.12*, were functional JA-Ile synthases. In addition, group I *GH3* family genes positively regulated resistance to the bacterial pathogen *Xoo* in rice mediated by JA homeostasis and the transcriptional patterns of JA-responsive genes [4]. In maize, *ZmGH3-2* and *ZmGH3-8* were highly up-regulated after *Colletotrichum graminicolum* infection, indicating that these two genes play essential roles in the *C. graminicolum* response [20]. In potato (*Solanum tuberosum* L.), the disruption of *StGH3.1* and *StGH3.5* genes could reduce the ability to resist *Ralstonia solanacearum* induction, thus making it more susceptible to *R. solanacearum* infection [21].

Sugarcane (*Saccharum* spp.) is the most important sugar and bioenergy crop in the world [22,23]. At present, sugarcane accounts for more than 85% of the total sugar production in China [24,25]. Therefore, the stable development of the sugarcane industry is of great significance to Chinese sugar supply safety [25,26]. Sugarcane smut, which is caused by *Sporisorium scitamineum*, is a particularly important fungal disease that seriously affects the yield and quality of sugarcane [27,28]. Cultivating smut-resistant sugarcane cultivars is the fundamental solution to the problem [27,28]. Hence, the identification and characterization of disease-resistance genes have profound meaning for molecular breeding in sugarcane, not only providing genetic resources but also laying a theoretical foundation. GH3 plays a crucial role in plant growth and development, and, in addition to the specific response to auxin, *GH3* genes are also transcriptionally regulated by ABA (abscisic acid), JA, SA, GA (gibberellins), BRs (brassinosteroids), ethylene, and biotic and abiotic stresses [4,11]. However, in sugarcane, the comprehensive information and expression patterns of GH3 family genes in the response to disease stresses are largely unknown, and the functional mechanism of the *GH3* gene has not yet been reported. The present study aims to set up a basis for the further study of the sugarcane *GH3* gene and provide certain candidate genes with potential applications in molecular breeding to improve crop disease resistance.

## 2. Results

### 2.1. Identification and Characterization of the SsGH3 Gene Family

A total of 41 *SsGH3* gene sequences were identified in the *Saccharum spontaneum* genome. The letters “a”, “b”, “c”, and “d” were designated for the genes with the same name that belong to the alleles, and the letter “e” and “f” were designated for the duplicated genes with the same name [29] (Appendix A in the Appendix A). Among these 41 *SsGH3* genes, the *SsGH3-3.1*, *SsGH3-1.2* and *SsGH3-1.3* genes had four alleles, the *SsGH3-2.6*, *SsGH3-2.7*, *SsGH3-2.2* and *SsGH3-2.4* genes contained three alleles, and the *SsGH3-1.1*, *SsGH3-1.4*, *SsGH3-2.1*, *SsGH3-2.3*, *SsGH3-2.5* and *SsGH3-2.8* genes had two alleles. Additionally, the *SsGH3-2.7e* and *SsGH3-2.7f* genes originated from dispersed duplications. The amino acid lengths of the 41 SsGH3 proteins were 510–759 aa, the molecular weights (MW) were 56.83–74.67 KDa, and the isoelectric points (pI) ranged from 5.01 to 7.91, of which only SsGH3-1.3b had a pI value greater than 7.0. In addition, the instability index ranged from 35.26 to 54.81, where the SsGH3-1.3a/1.3c/1.3d, SsGH3-1.4b, SsGH3-2.5b, SsGH3-2.7a, and SsGH3-2.8b proteins were presumed to be stable (instability index below 40), and the overall mean value of hydrophilicity was less than 0 (Appendix A in the Appendix A). Subcellular localization predicted that both group I and group II were localized on the chloroplast, while group III was localized on the cytoplasm and nucleus (Appendix A in the Appendix A). The secondary structure of the SsGH3 proteins consisted of the α-helix (32.94%~42.68%), β-turn (4.11%~6.72%), random curl (37.23%~44.14%), and extended chain (13.11%~18.63%), among which the α-helix and random curl are the main components (Appendix A in the Appendix A).

### 2.2. Phylogenetic Analysis Divided the SsGH3s into Three Sub-Groups

To better understand the evolutionary relationships of the *GH3* gene family in sugarcane and other species, we used the maximum likelihood (ML) method to generate a phylogenetic tree [9]. Twenty AtGH3 proteins [6], 12 OsGH3 proteins [8], 12 ZmGH3 proteins [10] and 41 SsGH3 proteins were roughly divided into three groups (I, II, and III) (Figure 1 and Appendix A in the Appendix A), which is the same as reported in the results of previous studies on the classification of GH3 protein families in *Arabidopsis* [30]. A total of 16 SsGH3 proteins were clustered into group I, including SsGH3-1.1a/1.1b, SsGH3-1.2a/1.2b/1.2c/1.2e, SsGH3-1.3a/1.3b/1.3c/1.3d, SsGH3-1.4a/1.4b, SsGH3-1.5, SsGH3-1.6, and SsGH3-1.7a/1.7b, which are involved in the JA signaling pathway and play a related regulatory function in the pathway (Figure 1). The SsGH3-2.1a/2.1b, SsGH3-2.2a/2.2b/2.2e, SsGH3-2.3a/2.3b, SsGH3-2.4a/2.4b/2.4c, SsGH3-2.5a/2.5b, SsGH3-2.6a/2.6b/2.6e, SsGH3-2.7a/2.7e/2.7f and SsGH3-2.8a/2.8b proteins were clustered into group II; presumably, they are involved in the regulation of growth hormones. The SsGH3-3.1a/3.1b/3.1c/3.1e and SsGH3-3.2 belonged to group III, part of a smaller branch of group II (Figure 1). Overall, members of the same family tend to have similar functions, so the grouping in the phylogenetic tree likely reflected the differentiation of gene function during evolution.

### 2.3. Phylogenetic, Conserved Motifs, and Gene Structures of the SsGH3 Family Genes

As shown in Figure 2 and Appendix A in the Appendix A, the SsGH3 proteins had a high level of conserved sequences. Among them, 97.56% of SsGH3 proteins contained 10 conserved motifs, with no difference in the order of arrangement. However, some conserved motifs were specific to the group classification. Motif 17 was only present in group I, and motif 14, motif 15, and motif 19 were absent from group I. In contrast, the remaining SsGH3s of both group II and group III contained motif 19, except for SsGH3-2.5a in group II and SsGH3-3.2 in group III. It is noteworthy that 41 *SsGH3* genes had intron numbers between 0 and 8, and the number of introns in group I varied widely, ranging from 2 to 8, while 15 *SsGH3* genes of group II and 5 *SsGH3* genes of group III had no introns. Generally, the numbers and lengths of exons and introns were specific for each group (Figure 2). The divergent exon-intron gene structures among the different phylogenetic subgroups suggested that those genes may evolve to accomplish diverse functions.

### 2.4. Chromosomal Location and Synteny Analysis of SsGH3 Gene Family

The results of chromosome localization showed that 41 *SsGH3* genes were unevenly distributed on 19 *S. spontaneum* chromosomes, among which Chr3A and Chr3D contained 5 *SsGH3* genes, Chr3B and Chr3C contained 4 *SsGH3* genes, Chr5B, Chr7A and Chr7D contained 3 *SsGH3* genes, Chr7B and Chr7B and Chr7C contained 2 *SsGH3* genes, and the rest of the chromosomes contained only one *SsGH3* gene (Appendix A in the Appendix A). In conclusion, most of the *SsGH3* genes were distributed on Chr3 and Chr7 (Appendix A in the Appendix A). To further explore the replication patterns of *SsGH3* genes in *S. spontaneum*, the results of the dispersed, proximal, segmental duplication/whole-genome duplication (WGD) and tandem duplication of *SsGH3* genes showed that among the 41 *SsGH3* genes, 24 (60.98%) WGD or segmental genes, 12 (29.27%) dispersed genes, 2 (4.88%) tandem genes, 1 (2.44%) proximal gene, and 1 (2.44%) singleton gene were detected (Figure 3B and Appendix A in the Appendix A). Therefore, it was hypothesized that the expansion of the *SsGH3* gene family was achieved mainly through genome-wide replication events (Figure 3B and Appendix A in the Appendix A). The gene covariation of the *SsGH3* gene family showed that there were 19 pairs of fragmentary duplicated genes in *S. spontaneum*; the Ka/Ks of these duplicate gene pairs were all less than 1.0, indicating that they were under purifying selection (Figure 3A and Appendix A in the Appendix A)

### 2.5. GO Annotation of SsGH3 Gene Family

To better analyze the biological processes involved in the SsGH3 protein in sugarcane, all *SsGH3* genes were functionally gene ontology (GO) annotated (Figure 4 and Appendix A in the Appendix A). The results showed that *SsGH3* genes were mainly enriched to molecular function, cellular component, and biological process (Figure 4 and Appendix A in the Appendix A). In the biological process, the *SsGH3* genes were mainly involved in abiotic stimulus (5.99%), radiation (5.99%), light stimulus (5.99%), and hormones (5.99%). In the cellular component, the *SsGH3* genes were enriched in organelles (62.64%), cytoplasm (15.66%), chloroplasts (10.84%), and plastids (10.84%), which may be related to the cellular localization and primary site of action of the SsGH3 protein. In terms of molecular function, the *SsGH3* genes were linked with acid-amino acid ligase activity (20%), indole-3-acetic acid amido synthetase activity (20%), ligase activity (40%) and catalytic activity (20%) (Figure 4 and Appendix A in the Appendix A). These results are consistent with previous studies, suggesting that the *SsGH3s* are widely involved in the stimulus response, growth and development, and metabolism.

### 2.6. Cis-Acting Regulatory Elements of SsGH3 Gene Family

In order to further understand the promoter *cis*-acting regulatory elements (CREs) of the *SsGH3* gene, we used the 2000 bp sequence upstream of the translation start site of the *SsGH3* gene family as the promoter sequence, and then submitted them to the PlantCARE database to predict the CREs. In total, we obtained 23 CREs, including phytohormone-responsive, growth and development-related, and stress-related elements. These were randomly distributed in the promoter sequences (Figure 5 and Appendix A in the Appendix A). The ABA-related elements (ABRE) were present in 85.37% of *SsGH3* gene promoters, and all *SsGH3* genes contain the methyl jasmonate (MeJA) response elements (TGACG-motif and CGTCA-motif), except for *SsGH3-2.3b* and *SsGH3-1.4b/1.4a*. However, the gibberellin response element (GARE-motif) was only contained in *SsGH3-2.8b/2.7e/2.4a*. In addition, another 80.49% of all *SsGH3* genes contained the *cis*-acting element involved in anaerobic induction (ARE), and among the growth and developmental response elements, 53.66% of *SsGH3* genes contain the *cis*-acting regulatory elements associated with meristematic tissue expression (CAT-box). These elements were present in the promoter part of the *SsGH3* gene family and occupy considerable quantitative distribution, implying that *SsGH3* genes are widely involved in plant responses to various stresses.

### 2.7. Expression Patterns of SsGH3 Gene Family, Based on RNA-Seq Databases

As shown in Figure 6A, in the transcriptome data analyses, the expression levels of genes in the three groups were significantly different. The expression level of the *SsGH3*-III group genes was relatively low in the five ROC22 tissues. Among the *SsGH3*-II group genes, the expression level of *SsGH3-2.2b/2.2e* in the five ROC22 tissues was much higher than that of all the other *SsGH3* genes, especially in the root, bud, and leaf, indicating that this gene may play a role in sugarcane growth and development. In *SsGH3*-I group genes, the expression level of *SsGH3-1.2e/1.2b/1.2c/1.2a* and *SsGH3-1.5/1.6* exhibited low or undetectable expression (FPKM < 1) in the five ROC22 tissues (Figure 6A).

The expression pattern of the *SsGH3* genes in two sugarcane cultivars ROC22 (smut-susceptible) and YC05-179 (smut-resistant) upon smut pathogen infection for 0–5 d was analyzed (Figure 6B). The *SsGH3-3.1a/3.1b/3.1c/3.1e/* and *SsGH3-3.2* genes in *SsGH3*-III group exhibited low expression levels (FPKM < 0.8) under *S. scitamineum* stress. The expression levels of *SsGH3*-I genes showed an up-regulated trend, except at individual stress time points. Furthermore, the expression level of *SsGH3-1.1a/1.4a/1.4b* genes in the YC05-179 was higher than ROC22, indicating that the regulatory pattern of these genes may play a positive role in the interaction between sugarcane and the smut pathogen. Meanwhile, *SsGH3-1.7a/1.7b/1.1b* genes were highly expressed in both YC05-179 and ROC22. In addition, the expression level of *SsGH3*-II genes in ROC22 was lower than YC05-179, while the expression pattern of *SsGH3-2.7a/2.8b* was not significantly different. These results indicated that the expression patterns of the *SsGH3* gene family in the infected sugarcane smut-resistant and -susceptible cultivars were different (Figure 6B).

### 2.8. Cloning and Sequence Analysis of the ScGH3-1 Gene

Most of the *SsGH3*-I group genes were gradually increased with the prolongation of the stress period during the interaction between sugarcane and the smut pathogen, suggesting that members of group I respond positively to smut pathogen infection. Furthermore, the cDNA sequence of the homologous gene of *SsGH3-1.7a/b*, belonging to group-I gene members, was successfully cloned from the sugarcane cultivar ROC22 leaves, namely *ScGH3-1* (GenBank accession number: OP429459). The ORF length of *ScGH3-1* was 1233 bp and encoded 410 amino acids (Appendix A in the Appendix A). The amino acid sequence similarity between *SsGH3-1.7a/1.7b* and *ScGH3-1* was as high as 99.51% and 93.47%, respectively (Appendix A in the Appendix A). Multiple sequence alignments revealed that ScGH3-1, SsGH3-1.7a/1.7b, AtGH3-11, OsGH3-5, and ZmGH3-11 had high similarity and belonged to group I (Figure 7 and Appendix A in the Appendix A). These six proteins also shared the conserved structures of α1, α6, β2-3, β7-9, β11-13. Meanwhile, Lys and Ser were present in each protein, among which ^267^DKN^269^ (Lys), a highly conserved amino acid, can interact with amino acid substrates, and ^152^YGA/S^154^ (Ser) can bind specifically to isoleucine (Figure 7).

### 2.9. ScGH3-1 Is a Nuclear and Membrane Localization Protein

Transient expression was performed in *N. benthamiana* leaves by an *Agrobacterium*-mediated method, with the empty vector pFAST-R05-*ScGH3-1*-GFP as the experimental group and the empty vector pFAST-R05-GFP as the negative control. In addition, H2B-mCherry [32] was used as a generalized nucleus stain and was performed in red fluorescence. The results indicated that the control group (35S::GFP) had no green fluorescence in the plasma membrane, cytoplasm, and nucleus of *N. benthamiana*. In contrast, for 35S::*ScGH3-1*::GFP in the experimental group, green fluorescence was observed in both the plasma membrane and nucleus of *N. benthamiana*. At the same time, H2B-mCherry could clearly be seen in the red fluorescence. Furthermore, merged fluorescence caused the overlap of red and green to form yellow light (Figure 8), indicating that the sugarcane S*c*GH3-1 was a nuclear and membrane localization protein.

### 2.10. ScGH3-1 Was Constitutively Expressed in Different Sugarcane Tissues

The relative expression level of the *ScGH3-1* gene in the root, bud, leaf, stem pith, and epidermis of 10-month-old ROC22 was detected by real-time quantitative PCR (RT-qPCR) (Figure 9). The results indicated that this gene was constitutively expressed in all these tissues. The lowest expression level of *ScGH3-1* occurred in the root and epidermis. The highest expression level of *ScGH3-1* occurred in the stem pith, while its relative expression level in the stem pith was 10.22- and 9.86-fold higher than that in the root and epidermis, respectively. Interestingly, the expression in the stem pith was 1.80- and 1.80-fold higher than that in the leaf and bud, respectively (Figure 9).

### 2.11. ScGH3-1 Is Involved in the Response to S. scitamineum, ABA, SA, and MeJA Stresses in Sugarcane

As shown in Figure 10A, after infection with *S. scitamineum* at 0 d to 3 d, the expression level of *ScGH3-1* in the sugarcane smut-resistant cultivar YT96-86 was significantly decreased but was significantly increased in the sugarcane smut-susceptible cultivar ROC22. Under 100 µM ABA stress, the gene expression increased 1.68-fold compared to the control (0 h) at 3 h, and then gradually decreased from 3 h to 24 h afterward (Figure 10B). Similarly, under 25 μM of MeJA stress for 3 h, the gene expression level of *ScGH3-1* was 1.41-fold higher than the control, following which point it dropped to the control level at 12 and 24 h (Figure 10C). After 5 mM salicylic acid (SA) treatment, the gene expression level of *ScGH3-1* was decreased from 0 to 3 h, while its transcript significantly increased from 3 to 12 h (Figure 10D). These results revealed that the *ScGH3-1* gene negatively responded to the stresses of *S. scitamineum* and the signaling molecules MeJA, SA, and ABA.

### 2.12. Transient Overexpression of ScGH3-1 Negatively Regulated the Defense Response of N. benthamiana to Pathogen Infection

After inoculation with *Fusarium solani* var. *coeruleum*, the 3,3-diaminobenzidine (DAB)-staining color in the leaves of *35S::ScGH3-1* was darker than the *35S::00* at 1 d and 6 d (Figure 11A). At the same time, after inoculation with *F. solanacearum* var. *coeruleum* at 6 d, *35S*::*ScGH3-1* leaves of *N. benthamiana* showed large yellow spots and curled leaf edges. (Figure 11A), the lesion areas of *35S::ScGH3-1* leaves were all significantly higher than those of the control (*35S::00*) (Figure 11B), which were 4.88-fold higher than the control. The RT-qPCR analysis showed that the expression levels of the HR marker gene (*NtHSR201*), SA-related gene (*NtPR3)*, ethylene synthesis (ET)-dependent gene (*NtAccdeaminase)*, and JA-related genes (*NtLOX1* and *NtDEF1*) were significantly down-regulated in *35S*::*ScGH3-1* leaves at 1 d (Figure 11C), and were 0.09-, 0.21-, 0.20-, 0.18- and 0.15-fold that of the control, respectively. After inoculation with *F*. *solani* var. *coeruleum* over 6 d, the expression levels of *NtHSR201*, *NtDEF1*, *NtLOX1* and *NtPR3* had all decreased, and were 0.09-, 0.30-, 0.27-, and 0.39-fold that of the control, respectively (Figure 11C).

When inoculated with *Ralstonia solanacearum* for 1 day, both the leaves of *35S::00* and *35S::ScGH3-1* were slightly shrunken, and there was no significant difference in DAB staining color (Figure 11D). However, the disease symptoms in the leaves of *35S::ScGH3-1* were shown to be more serious than those of the control at 6 d. In addition, after being inoculated with *Ralstonia solanacearum* for 6 days, the DA-staining color in the leaves of *35S::ScGH3-1* was darker than in the control (Figure 11D), and the lesion areas of *35S::ScGH3-1* leaves were all significantly higher than those of the control, at 5.19-fold higher than those of the control (Figure 11E). RT-qPCR analysis showed that, the expression levels of *NtHSR201*, *NtAccdeaminas*, *NtLOX1*, *NtDEF1* and *NtPR3* were significantly down-regulated after 6 d in *35S::ScGH3-1* at 0.49-, 0.22-, 0.36-, 0.16-, and 0.36-fold those of the control, respectively (Figure 11F).

## 3. Discussion

In the present study, the bioinformatics method was used to analyze the evolutionary and functional grouping of the *SsGH3* gene family, based on the *S. spontaneum* genome, and all the SsGH3 proteins identified in this study were classified into three groups (I, II, and III) (Figure 1). Consistent with our findings, Khan et al. [30] classified the *Arabidopsis* GH3 proteins into three classes, according to their functional properties and sequence similarity, which was also in accordance with the classification of GH3 proteins in maize [10], rice [33], wheat [9], and potato [21]. Staswick et al. [34] found that several GH3 proteins in *Arabidopsis* had the function of catalyzing the adenylation of SA, IAA, or JA. Studies on the function of related GH3 proteins in *Arabidopsis* and rice indicated that each species contained group I, which was mainly associated with JA signaling [33,35]. In *Arabidopsis*, AtGH3-11 (JAR1/FIN219) and AtGH3-10 (DFL2) adenylated JA and catalyzed the formation of JA-Ile, and AtJAR1 catalyzed the synthesis of JA with the ethylene precursor 1-aminocyclopropane-1-carboxylic acid (ACC), which regulated the synthesis of JA and ethylene [35]. In potatoes, *StGH3.1* and *StGH3.5* in group I induced JA to JA-IIe through the JA signaling pathway, thereby affecting plant disease resistance [21]. We thus speculate that Group I of the *SsGH3* gene family may also be related to JA signaling. Besides, in this study, group II had the largest number of GH3 proteins and was associated with growth hormone regulation. In previous studies, group II *GH3* genes in rice, including *OsGH3-1*, *OsGH3-2*, *OsGH3-8*, and *OsGH3-13* were found to be involved in growth hormone regulation [19]. Similarly, the group II *GH3* genes in *Arabidopsis AtGH3.1*, *AtGH3.2* (*YDK*), *AtGH3.3*, *AtGH3.4*, *AtGH3.5* (*AtGH3a*), *AtGH3.6* (*DFL1*), *AtGH3.9*, and *AtGH3.17* catalyzed the adenylation or linkage of IAA to amino acids, while *AtGH3.5* also catalyzed the adenylation and linkage of SA to amino acids [36]. In contrast, group III had far fewer proteins than subfamilies I and II [36]. There were 10 members of *Arabidopsis* class-III GH3 proteins, among which, AtGH3.12 (PBS3) was thought to catalyze the linkage of SA to amino acids [36]. The above studies suggested that the *GH3* gene family has function differentiation.

The types and numbers of motifs are an important basis for GH3 family classification. All SsGH3 proteins had conserved GH3 structural domains (Motif 1 to Motif 14) and most SsGH3 family members in the same category had similar motif types and numbers (Figure 2); presumably, *SsGH3* genes may have different biological functions, depending on the types and numbers of motifs. Differences between exon-intron structures of gene family members are also considered tools for phylogenetic grouping and this diversity is an important component of gene family progression, diversification, and novel functions [37,38,39]. In the present study, there were obvious differences in the numbers of exons/introns of *SsGH3* genes belonging to the same group (Figure 2), and it is hypothesized that these differences in gene structure led to the functional differentiation of *SsGH3* genes. In plants, JA, SA, and ABA play important roles involved in both biotic and abiotic stresses. Liu et al. [40] demonstrated that the soybean *GH3* gene promoter contains at least three growth hormone-response elements (AuxREs), of which D1 (25 bp) and D4 (25 bp) are contained in a 76-bp sequence range, and both contained the conserved TGTCTC sequence, which is required for growth hormone response. In the present study, only the *SsGH3-2.7e, SsGH3-3.2, SsGH3-1.3a*, and *SsGH3-1.5* genes contained the AuxRE or AuxRR-core elements, but 38 out of 41 *SsGH3* genes contained MeJA-related elements (CGTCA-motif and TGACG-motif) and all of the *SsGH3*-III-group genes contained the SA-related elements (TCA-element) (Figure 5). In addition, the ABA-related elements (ABRE) were found in 85.73% (35 out of 41) of the *SsGH3* gene family (Figure 5), which suggests that the function of the *SsGH3* gene family is mainly involved in the JA, SA, and ABA signaling pathways. Previous studies showed that the rice GH3 protein (OsGH3-5) was predicted to be expressed in the endoplasmic reticulum [41], while AtGH3.11 in *Arabidopsis* was located in the nucleus, cytoplasm, and vesicles [42]. In this study, the ScGH3-1 protein was found to be expressed in the cell membrane and nucleus via signaling assays in tobacco cells (Figure 8); the results may suggest that the peroxisome on the nuclear membrane will release JA after the cell is stressed [43], the ScGH3-1 protein will rapidly bind to it, and the formed JA-Ile coupling will directly enter the nucleus from the nuclear pore, to act on the receptor protein SCF^COI1^ [44].

GH3 members of the different species have tissue expression specificity. In potato plants, Zhang et al. [21] found that the *GH3* gene family was not highly expressed in the underground parts (stolons and tubers), while their expression in the aboveground parts was mainly present in the different parts and periods of flowering organs and fruit. The *TaGH3* genes of wheat were found to be expressed mainly in the roots and leaves, only in different amounts [9], while the *OsGH3.2* of rice [45] was found to be the gene most highly expressed in roots. In the present study, most *SsGH3* genes of group I and group II were constitutively expressed in five tissues (the root, bud, leaf, stem pith, and epidermis) of ROC22, whereas the expression of group III was low in the five tissues (Figure 6A). Interestingly, as the expression patterns were identical, it is speculated that *SsGH3* genes in the same group may perform the same function during sugarcane growth and development (Figure 6A). In addition, the *ScGH3-1* gene was also constitutively expressed in all five tissues, and the highest expression was found in the stem pith (Figure 9). The *GH3* genes have been reported to mediate plant defense responses and adaptations to biotic and abiotic stresses [19,20,21]. The overexpression of *GH3.5* in *Arabidopsis* caused further SA accumulation, which enhanced the ability of plants to resist external stresses [16]. Meanwhile, Ding et al. [18] found that the overexpression of *GH3-8* resulted in enhanced resistance to the rice pathogen *Xoo*. Sugarcane smut is the most damaging fungal disease in sugarcane planting areas in China. In the present study, after inoculation with the smut pathogen, the expression level of *ScGH3-1* was significantly decreased in the smut-resistant cultivar YT96-86, while it was significantly increased in the disease-susceptible cultivar ROC22 (Figure 10A). The results indicated that S*cGH3-1* may play a negative regulatory role in the process of sugarcane resistance to smut pathogen stress, which was different from the positive regulation of *GH3* genes in rice [18,19], *Arabidopsis* [12,13,14], and potato [21]. In addition, to further verify whether *ScGH3-1* can cause the accumulation of JA and SA, we found that the expression level of *ScGH3-1* was elevated after the application of exogenous ABA, MeJA, and SA (Figure 10B–D), suggesting that *ScGH3-1* participates in the plant stress response through the ABA, JA, and SA pathway, which is consistent with the findings that treatment with ET [46], JA [47], SA, ABA, and pathogen infestation [16] induced the up-regulation of *GH3* gene expression.

The *GH3* gene plays an important role in the plant-pathogen interaction system. The maize *GH3* gene played a role in immune resistance to maize pathogens by participating in the SA signaling pathway [20]. The *GH3.3*, *GH3.5*, *GH3.6*, and *GH3.12* genes in rice possessed JA-Ile synthase activity by converting free JA to the active form of JA-Ile, thereby being resistant to rice bacterial diseases [19]. Studies on *Medicago truncatula* and *Pinus pinaster* also found that the *GH3* family genes were involved in the regulation of plant response to mycorrhizal infestation processes [48,49]. Interestingly, our study found that after inoculation with *F. solanacearum* var. *coeruleum* and *R. solanacearum*, the expression of JA, SA, ET, and HR-related pathway genes was down-regulated in tobacco that was transiently over-expressed with *ScGH3-1* (Figure 11), indicating that the *ScGH3-1* gene may be involved in the JA, SA, ET, and HR pathways as a negative regulatory gene to improve the resistance to *F. solanacearum* var. *coeruleum* and *R. solanacearum*.

In summary, *ScGH3-1* is a negative regulator by means of mediating the JA, SA, ET, and HR pathways. Combining the results of phenotypic observations, DAB staining, and immune-related gene expressions, a proposed model was drawn up for *ScGH3-1*-mediated regulation in the response of *N. benthamiana* plants to pathogen infection, which indicated that the transgenic overexpression of the *ScGH3-1* gene in *N. benthamiana* decreased pathogen tolerance by inhibiting the JA signaling pathway to reduce the expression of the HR marker gene, *NtHSR201*, SA-related gene, *NtPR3*, ET-dependent gene, *NtAccdeaminase*, and JA-related genes, *NtLOX1* and *NtDEF1*, and the physiological and biochemical levels of the plants were altered at the same time (Figure 12).

## 4. Materials and Methods

### 4.1. Plant Materials and Treatments

The sugarcane cultivars YT96-86 (smut-resistant) and ROC22 (smut-susceptible) were provided by the Fujian Key Laboratory of Sugarcane Biology and Genetic Breeding, Ministry of Agriculture and Rural Affairs. Healthy and uniformly growing sugarcane stems at the 10-month-old stage were selected and germinated at 32 °C. When the buds grew to 1~2 cm, they were inoculated with 5 × 10^6^ units/mL (containing 0.01% Tween-20, *v*/*v*) [50] of *S. scitamineum*-mixed spore suspension, and the control was inoculated with sterile water. All treatments were incubated at 28 °C for 16 h in the light and 8 h in the dark. The sugarcane shoots were cut and frozen in liquid nitrogen at 0 h, 6 h, 24 h, and 48 h after inoculation. Three biological replicates were set up, each sample contained three single shoots, and the materials were used for the RT-qPCR analysis of gene expression in response to sugarcane smut stress.

In terms of the different sugarcane tissues, white young roots, leaf (+1), epidermis (+1), buds (6th–8th from the base), and stem pith (6th and 7th from the base) were sampled from nine healthy and mature 10-month-old ROC22 plants [51]. Three biological replicates were set up at each time point and the materials were used for the RT-qPCR analysis of gene expression.

The leaves of four-month-old ROC22 tissue culture seedlings were sprayed with 100 µM ABA, 25 µM MeJA (containing 0.1% ethanol and 0.05% Tween-20, *v*/*v*) and 5 mM SA (containing 0.01% Tween-20, *v*/*v*) [51,52], respectively. Three biological replicates were set up at each time point and the materials were used for the RT-qPCR analysis of gene expression under ABA, MeJA, and SA stresses.

### 4.2. Identification of the SsGH3 Gene Family

Twenty *Arabidopsis* GH3s (AtGH3s) from the *Arabidopsis* Information Resource (TAIR10) database (https://www.arabidopsis.org/index.jsp, accessed on 3 March 2022), 12 maize GH3s (ZmGH3s) from the Maize Genetics and Genomics Database (Maize GDB) (https://www.maizegdb.org/, accessed on 3 March 2022) and 12 rice GH3s (OsGH3s) protein sequences from the Rice Genome Annotation Project (RGAP) database (https://rice.plantbiology.msu.edu/, accessed on 3 March 2022) were retrieved and downloaded, respectively. In order to identify the SsGH3 protein sequences in *Saccharum.* spp, we used the genome of the sugarcane ancestor, *S. spontaneum* AP85-441 (http://www.life.illinois.edu/ming/downloads/Spontaneum.genome/, accessed on 3 March 2022) [29]. Then, we used a profiled hidden Markov model (HMM), implemented with default parameters in HMMER v3.2.1 [53], to search for SsGH3 proteins with the GH3 domain (PF 03321) in the Pfam (http://pfam.xfam.org/, accessed on 16 March 2022) [54] database. Finally, we screened the gene sequences with intact conserved GH3 structural domains, based on the structural features of *GH3* family genes.

### 4.3. Conserved Motifs and Gene Structures of the SsGH3 Gene Family

In the prediction of conserved SsGH3 protein motifs by the MEME online software (http://meme-suite.org/tools/meme, accessed on 16 March 2022), the number of different motifs was 20, and the motif length ranged from 6 to 50 amino acids [55]. The structural information of exon-intron was extracted from a GFF3 file in the *S. spontaneum* genome data. Conserved protein motifs and gene structures were visualized using TBtools [31].

### 4.4. Sequence Features, Gene Duplications, and the Synteny of SsGH3 Family Genes

The characteristics of SsGH3s protein length, molecular weight (MW), isoelectric point (pI), instability index, and grand average hydropathicity (GRAVY) were predicted by the ExPASy server10 (https://prosite.expasy.org/, accessed on 21 March 2022) [56]. The Plant-mPLoc (https://www.csbio.sjtu.edu.cn/cgi-bin/Plant, accessed on 23 March 2022) was used to predict the subcellular localization of SsGH3s. The duplication pattern and synteny of the *SsGH3* genes was analyzed using the Multiple Collinearity Scan toolkit (MCScanX) [57]. Ka/Ks values between orthologous gene pairs were calculated using TBtools [31].

### 4.5. GO Annotation of SsGH3 Family Genes

The GO annotation of SsGH3 protein sequences was performed using Blast2GO v5, with the parameters set to default parameters [58]. The target sequences were identified by screening, after sequence comparison using BLASTP, and then analyzed and annotated by the InterProScan (https://www.ebi.ac.uk/interpro, accessed on 25 March 2022) program. Finally, the target sequences were subdivided into three categories, biological processes, cellular components, and metabolic pathways by GO annotation.

### 4.6. Cis-Acting Regulatory Elements in the Promoter Regions of SsGH3 Family Genes

Based on the genome of *S. spontaneum* [29], the promoter regions 2 kb of genomic DNA sequence upstream of the translation start site was used as the promoter sequence and was accessed through the PlantCARE online program (http://bioinformatics.psb.ugent.be/webtools/plantcare/html/, accessed on 27 March 2022) for predicting *Cis*-acting regulatory elements [59]. Then, the results were visualized using TBtools [31].

### 4.7. Expression Patterns of SsGH3 Family Genes, Based on the Transcriptome Databases

The RNA-Seq data of 5 different tissues (root, epidermis, pith, leaf, and bud) of *Saccharum* hybrid cultivar ROC22 (unpublished) were used to analyze the tissue expression patterns of the *SsGH3* gene family in *Saccharum* [60]. Three biological replicates were set up, each containing three biological replicates, and the material was used to analyze the tissue-specific expression of the *ScGH3-1* gene. The RNA-Seq data of YC05-179 (smut-resistant) and ROC22 (smut-susceptible) cultivars infected by *S. scitamineum* for 0, 1, 2, and 5 days were used to analyze the expression patterns of the *SsGH3* gene family under *S. scitamineum* stress [61]. FPKM was used as a measure of the transcript or gene expression level to convert the expression of the *ScGH3-1* gene by log_2_ (FPKM); the heat map was constructed using TBtools [31].

### 4.8. Gene Cloning and Bioinformatics Analysis of Sugarcane ScGH3-1 Gene

RNA was extracted from ROC22 leaves after 7 d of sugarcane smut fungus infestation using the Trizol (Invitrogen, Carlsbad, CA, USA) method, then reverse-transcribed into first-strand cDNA using SuperScript IV reverse transcriptase (Fermentas, Shanghai, China) as a template for cloning the sugarcane *ScGH3-1* gene. The primers were designed using Primer 5 software (Appendix A in the Appendix A). The PCR procedure was as follows: pre-denaturation at 95 °C for 3 min; denaturation at 98 °C for 10 s, annealing at 53 °C for 15 s, extension at 72 °C for 30 s, for 35 cycles; final extension at 72 °C for 7 min. The PCR product was ligated to the cloning Blunt-Zero vector (Transgen Biotech, Beijing, China) and was then transformed into *Escherichia coli* DH5α; the PCR products of Blunt-Zero-*ScGH3-1* were recovered for gel detection and sent to Fuzhou Shangya Biotechnology Co., Ltd. for sequencing. The *ScGH3-1* gene ORF was predicted by the ORF Finder (https://www.ncbi.nlm.nih.gov/orffinder/, accessed on 13 May 2022) and the conserved domain was predicted by the conserved domains database (CDD) (http://www.ncbi.nlm.nih.gov/Structure/cdd/wrpsb.cgi, accessed on 13 May 2022). The Clustal Omega program (https://www.ebi.ac.uk/Tools/msa/clustalo/, accessed on 13 May 2022) and the ESPrit3.0 online software (http://espript.ibcp.fr/ESPript/cgi-bin/ESPript.cgi, accessed on 13 May 2022) were used for multiple alignments of the group I amino acid sequences of GH3-1 proteins. The pI and MW of this protein were predicted by the ExPASy online software (https://web.expasy.org, accessed on 13 May 2022) [56].

### 4.9. Subcellular Localization of Sugarcane ScGH3-1 Gene

The gateway introductory vector primers (Appendix A in the Appendix A) were designed according to the sequence of the *ScGH3-1* gene and the pEASY-Blunt Zero-*ScGH3-1* plasmid was used as a template for PCR amplification. Then, the gel containing the target DNA fragment was recovered and purified, and the *ScGH3-1* gene (without the stop codon) was ligated to the introductory vector, pDONR-207. The target gene was then constructed into the subcellular localization vector, pFAST-R05-GFP. The ligated product was transformed into *E. coli* DH5α, single clones were picked, and the bacterial broth was detected by PCR and validated by sequencing. The positive recombinant introductory vector plasmids were identified as pDONR-207-*ScGH3-1* and the positive recombinant subcellular localization vector plasmids were identified as pFAST-R05-*ScGH3-1*-GFP. The pFAST-R05-*ScGH3-1*-GFP plasmids were transformed into *A*. *tumefaciens* GV3101 and the monoclonal clones were picked and tested via PCR in bacterial broth. After detection, the clones were cultured in LB liquid medium containing 50 μg/mL rifampicin and 50 μg/mL kanamycin sulfate. Then, the bacterial solution concentration was adjusted to OD600 ≈ 0.6 with 10 mM magnesium chloride (MgCl_2_), an infiltration solution containing 10 mM MES and 200 μM acetosyringone, and 6-leaf-old *N*. *benthamiana* plants were selected for leaf injection [62]. The control group was *A. tumefaciens* GV3101, containing empty pFAST-R05-GFP. After injection, the clones were incubated for 2 d at 28 °C for 16 h in the light/8 h in darkness. In addition, H2B-mCherry was used for nuclear staining [32]. Then, the clones were examined with a LEICA TCS SP8 laser confocal microscope (Leica, Wetzlar, Germany) to observe the subcellular localization results.

### 4.10. Quantification of the Sugarcane ScGH3-1 Gene by RT-qPCR Analysis

Primers for RT-qPCR detection of the *ScGH3-1* gene were designed using IDT online software (https://sg.idtdna.com/PrimerQuest, accessed on 15 June 2022) (Appendix A in the Appendix A). GAPDH (GenBank accession number CA254672) was selected as the internal reference gene (Appendix A in the Appendix A) [63]. The tissue-specific expression of the *ScGH3-1* gene and its expression levels under smut pathogen, MeJA, SA, and ABA stresses were examined by RT-qPCR using the Applied Biosystems QuantStudio 3 real-time quantitative PCR system (Thermo Fisher, Waltham, USA). The RT-qPCR reaction system was prepared according to the instructions for the 2 × ChamQ Universal SYBR qPCR Master Mix [64,65]. The program was adopted as the default program of Applied Biosystems: “50 °C, 2 min; 95 °C, 10 min; 95 °C, 15 s, 60 °C, 1 min, 40 cycles”. The relative gene expression was first calculated using 2^−ΔΔCt^ [66]; then, the significance level of the experimental data was analyzed using DPS 9.50 software, and finally, graphs were plotted using Origin 2021 (OriginLab Corporation, Northampton, MA, USA). All data points were means ± standard deviation (*n* = 3). Bars superscripted by different lower-case letters indicate significant differences, as determined by Duncan’s new multiple-range test (*p*-value < 0.05).

### 4.11. Transient Expression of the ScGH3-1 Gene in N. benthamiana

The transient overexpression of pEarleyGate 203 and pEarleyGate 203-*ScGH3-1* in the 7-leaf-old *N. benthamiana* plant that was inoculated with *R. solanacearum* and *F. solani* var. *coeruleum* for one day [67,68,69], respectively. *N. benthamiana* was incubated at 28 °C for 16 h in light/8 h in darkness and was subsequently followed for leaf phenotypic alterations. Three biological replicates were set up. Two leaves of *N. benthamiana* tobacco were collected at 1 d and 6 d after inoculation; one was stained with the DAB histochemical to detect hydrogen peroxide (H_2_O_2_) accumulation, while the other was stained with RT-qPCR to analyze the expression of tobacco immune-related marker genes [68]. These immune-related marker genes included the HR marker gene, *NtHSR201* [68], the SA-related gene, *NtPR3* [69], the ET-dependent gene, *NtAccdeaminase* [70], and the JA-related genes, *NtLOX1* and *NtDEF1* [70] (Appendix A in the Appendix A). *NtEF-1α* was used as an internal reference gene (Appendix A in the Appendix A).

## 5. Conclusions

In the present study, 41 *SsGH3* family genes were identified and classified into three groups (I, II, and III). These 41 genes were distributed on 19 *S. spontaneum* chromosomes, mainly concentrated on Chr3 and Chr7. Synteny analysis showed that the *SsGH3* family genes had 19 pairs of fragmentary duplicated genes in *S. spontaneum*; the Ka/Ks of these duplicate gene pairs were all less than 1.0. The GO annotation indicated that the *SsGH3* family genes were mainly involved in phytohormone response, defense, growth and development, and metabolism. The *ScGH3-1* gene was cloned from the sugarcane cultivar ROC22; it was constitutively expressed in the root, bud, leaf, stem pith, and epidermis, with the highest expression in the stem pith. After inoculation with *S. scitamineum*, the *ScGH3-1* gene was up-regulated in ROC22 but down-regulated in the YT96-86. In addition, the *ScGH3-1* gene showed an up-regulated expression under ABA, MeJA, and SA treatments. Compared with the control, the *N. benthamiana* that transiently overexpressed the *ScGH3-1* gene demonstrated high susceptibility after inoculation with both *F. solanacearum* var. *coeruleum* and *R. solanacearum*. Expression analysis of the tobacco immune-related genes revealed that the *ScGH3-1* gene may be involved in the JA, SA, ET, and HR pathways as a negative regulatory gene to improve the resistance to *F. solanacearum* var. *coeruleum* and *R. solanacearum*. Finally, a potential model for the *ScGH3-1*-mediated regulation of resistance to pathogen infection in transgenic *N. benthamiana* plants was proposed. This study aims to set up a basis for the further study of the sugarcane *GH3* gene and provide some candidate genes, with potential application in molecular breeding to improve crop disease resistance.

## Figures and Tables

**Figure 1 ijms-23-12750-f001:**
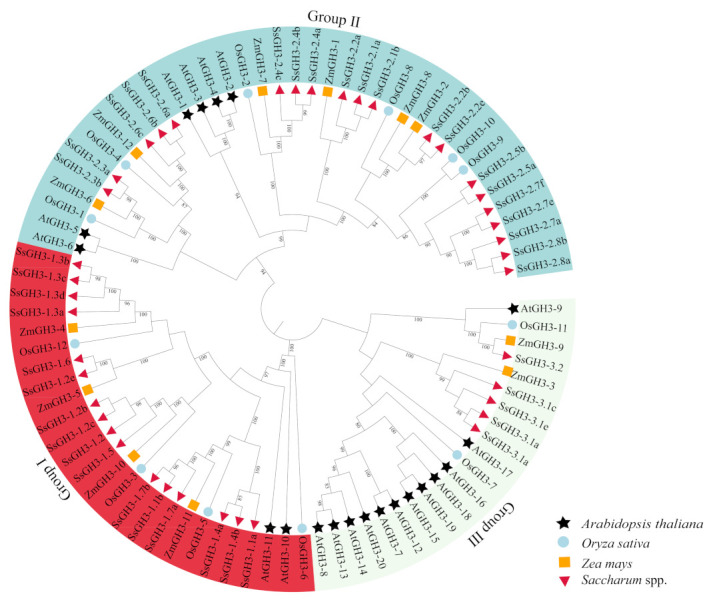
Phylogenetic analysis of the GH3 homolog proteins from 4 plant species. MEGA X with the maximum likelihood (ML) method was used to create the phylogenetic tree. GH3-related proteins from four species: *Saccharum spontaneum* (Ss), *Oryza sativa* (Os), *Zea mays* (Zm), and *Arabidopsis thaliana* (At). All the GenBank accession numbers are listed in Appendix A in the Appendix A. Three major clades were distinguished with three colors, and GH3 proteins from different plant species were decorated with different colors and shapes.

**Figure 2 ijms-23-12750-f002:**
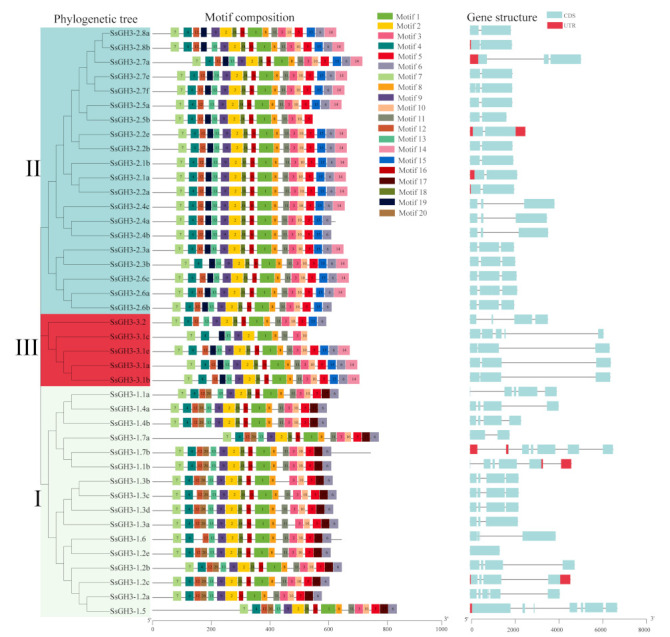
Phylogenetic tree, conserved motifs, and gene structure of the *SsGH3* gene family. Different colors and Latin numbers (I, II and III) on the phylogenetic tree represent three subgroups of the *SsGH3* genes. In the motif pattern, the 20 conserved motifs are displayed in different colored boxes. In exon-intron analysis, the grey line represents introns, the blue box represents the coding sequence, and the red box represents the non-coding sequence. I, II and III represented different subgroups.

**Figure 3 ijms-23-12750-f003:**
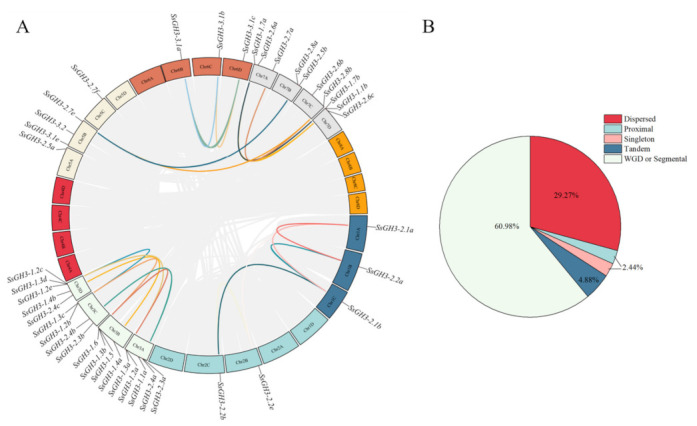
Synteny analysis (**A**) and expansion mechanisms (**B**) of the *SsGH3* family gene. The grey lines represent all the genes that have been replicated in the genome; the different colored lines represent the *SsGH3* genes that have been replicated. Different colors in the pie chart represent different types of gene duplication. WGD or segmental, whole-genome duplication/segmental duplication; Dispersed, dispersed duplication; Proximal, proximal duplication; Tandem, tandem duplication; Singleton, singleton duplication.

**Figure 4 ijms-23-12750-f004:**
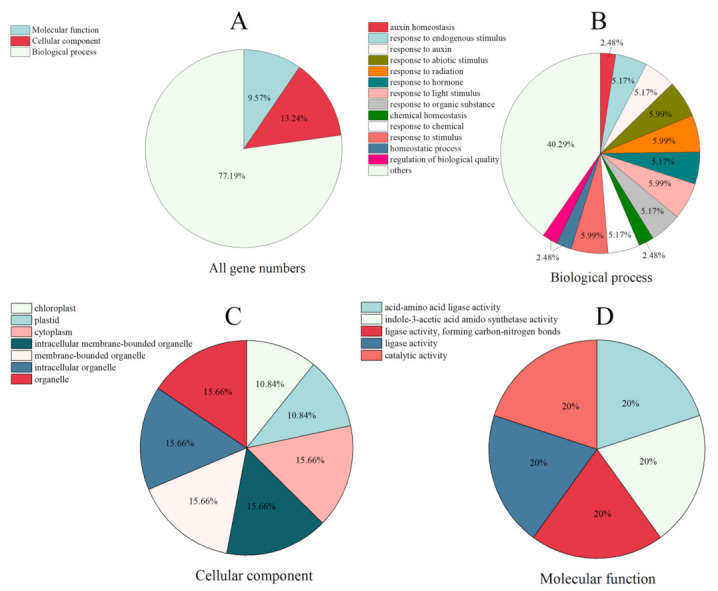
GO annotation of the *SsGH3* family genes. GO terms are summarized in three main categories. (**A**) All GO terms in three main categories. (**B**–**D**) The functional annotation analysis of *SsGH3* genes in terms of biological process, cellular component, and molecular function, respectively.

**Figure 5 ijms-23-12750-f005:**
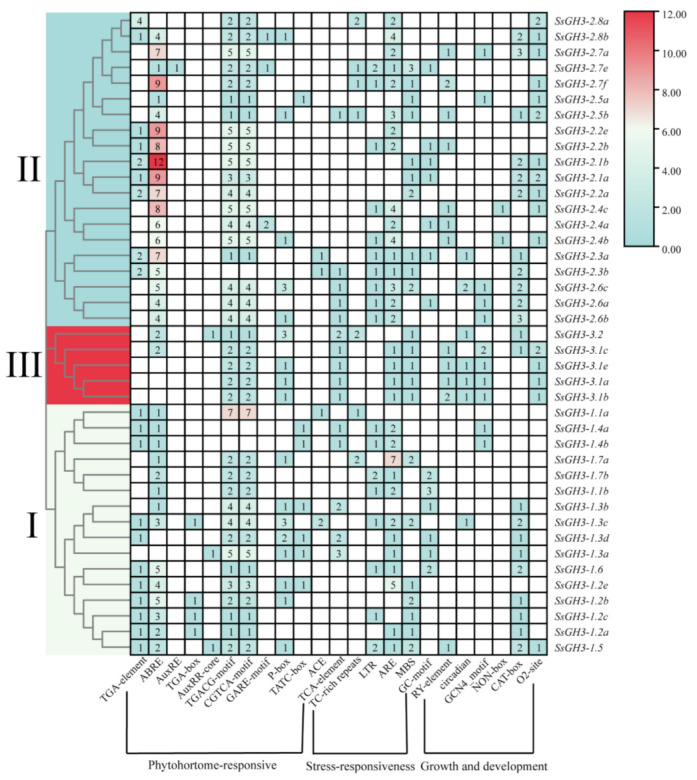
*Cis*-regulatory element analysis of the *SsGH3* family genes. Different colors and Latin numbers (I, II and III) on the phylogenetic tree represent three subgroups of the *SsGH3* genes. The number of each *cis*-regulatory element is shown in the heatmap box ranging from blue to red, with the white boxes indicating no corresponding *cis*-regulatory element.

**Figure 6 ijms-23-12750-f006:**
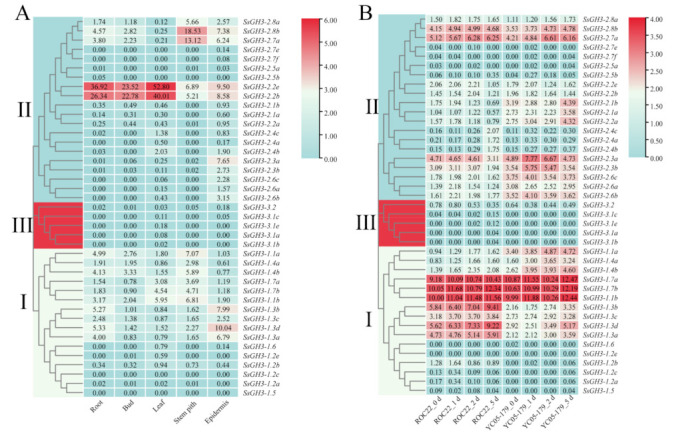
Expression patterns of the *SsGH3* gene family in sugarcane, based on RNA-seq. (**A**) Tissue expression patterns of the *SsGH3* gene family in the sugarcane cultivar ROC22. (**B**) Expression patterns of *SsGH3* gene family in the interaction between different sugarcane genotypes and *S. scitamineum*. ROC22_0 d/1 d/2 d/5 d and YC05–179_0 d/1 d/2 d/5 d represent the sugarcane smut-susceptible cultivar ROC22 and smut-resistant cultivar YC05–179 under *S*. *scitamineum* treatment for 0 d, 1 d, 2 d, and 5 d, respectively. The number on the box represented the FPKM (number of fragments per kilobase of transcript per million mapped) value, and the expression level of *SsGH3* genes was transformed by log_2_ (FPKM). The blue area in the figure indicates that the FPKM value was 0. The clustering tree on the left side of the figure was constructed using the k-nearest neighbor method, and different colors and Latin numbers (I, II and III) on the phylogenetic tree represent three subgroups of the *SsGH3* genes. The heat map was constructed with TBtools [31].

**Figure 7 ijms-23-12750-f007:**
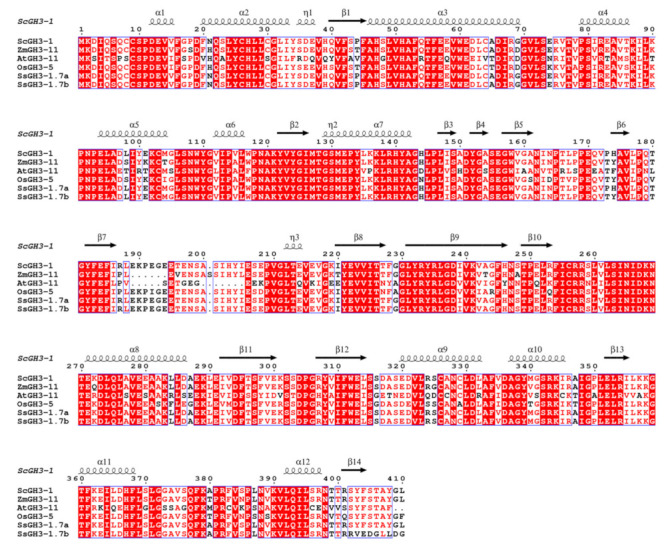
Multiple alignments of group-I amino acid sequences of GH3-1 proteins for *S. spontaneum* (SsGH3-1.7a/1.7b), *A. thaliana* (AtGH3-11), *O. sativa* (OsGH3-5) and *Z. mays* (ZmGH3-11). Conserved residues are highlighted with red boxes, with similar residues shown in a lighter color.

**Figure 8 ijms-23-12750-f008:**
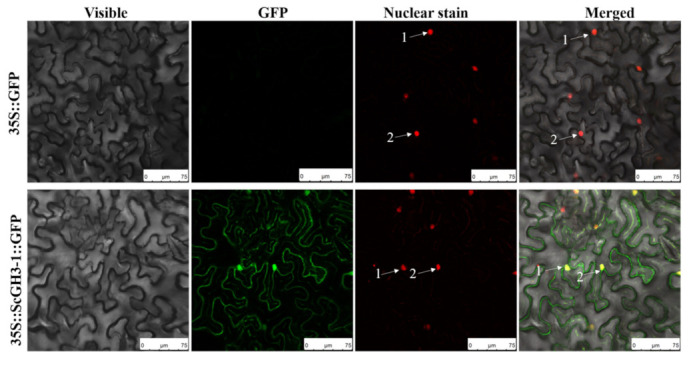
Subcellular localization of ScGH3-1 protein in *N. benthamiana*. The images were taken from three fields of view, including visible light, green fluorescence, red fluorescence, and merged green fluorescence. Here, 35S::GFP, the *Agrobacterium tumefaciens* strain, carried the empty vector pFAST-R05-GFP; 35S::ScGH3-1::GFP, the *A. tumefaciens strain*, carried the recombinant vector pFAST-R05-ScGH3-1-GFP; white arrows indicate the nucleus. H2B-mCherry is used for nuclear staining. Bar = 75 μm.

**Figure 9 ijms-23-12750-f009:**
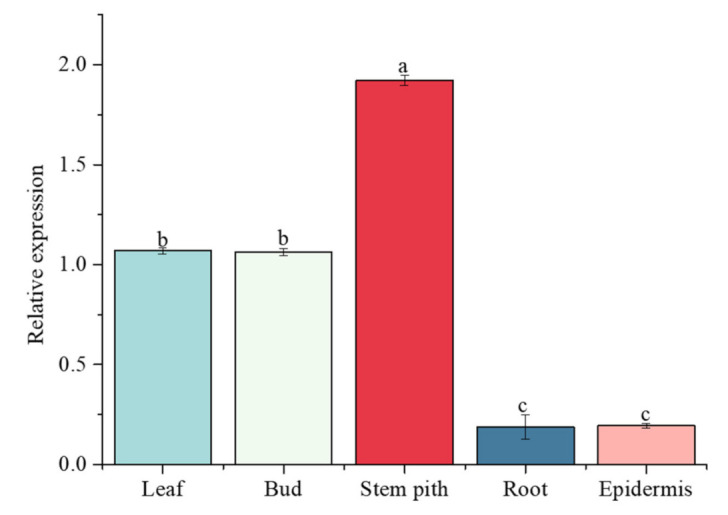
Expression of ScGH3-1 genes in the different tissues of sugarcane ROC22. Other sugarcane tissue data are normalized with sugarcane root. All the data points are means ± standard deviation (*n* = 3). Bars superscripted by different lower-case letters indicate significant differences, as determined by Duncan’s new multiple range test (*p*-value < 0.05).

**Figure 10 ijms-23-12750-f010:**
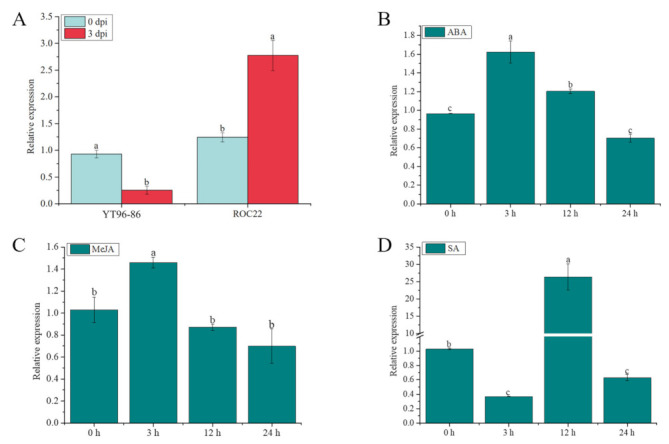
Expression of the ScGH3-1 gene under *S. scitamineum*, ABA, SA, and MeJA stresses. (**A**) Expression of ScGH3-1 gene in the interaction between sugarcane and *S. scitamineum*. (**B**–**D**) Expression of ScGH3-1 gene under ABA, MeJA, and SA stresses. Data were normalized to the glyceraldehyde-3-phosphate dehydrogenase (GAPDH) expression level. All the data points are means ± standard deviation (*n* = 3). Bars superscripted by different lower-case letters indicate significant differences, as determined by Duncan’s new multiple-range test (*p*-value < 0.05).

**Figure 11 ijms-23-12750-f011:**
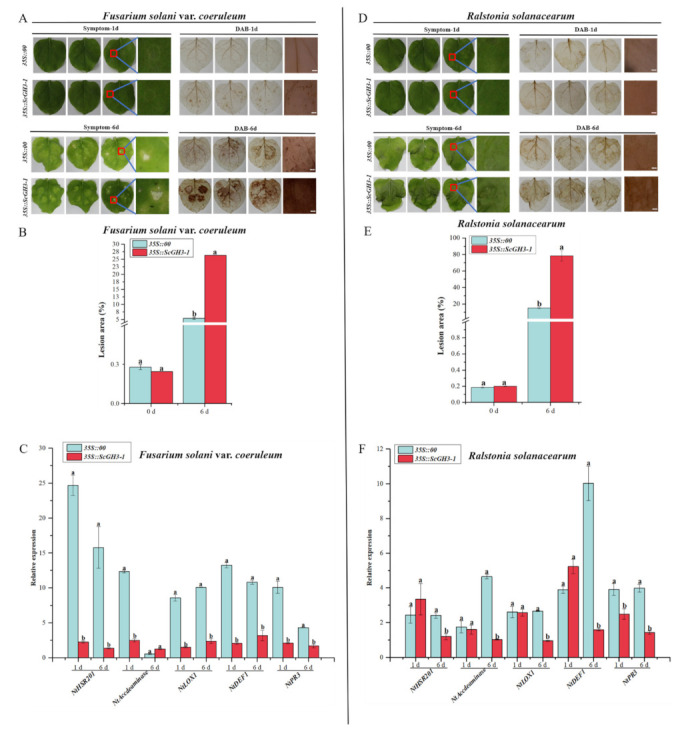
Resistance effect of the transient overexpression of *ScGH3-1* in *N. benthamiana* leaves. (**A**–**D**) Phenotype and DAB staining of *N. benthamiana* leaves after inoculation with *Ralstonia solanacearum* and *Fusarium solani* var. *coeruleum* for 1 d and 6 d, bar = 2 mm. (**B**,**E**) Lesion area of leaves after infection with *R. solanacearum* and *F. solanacearum* var. *coeruleum* for 1 d and 6 d. (**C**,**F**) Expression analysis of the immunity-related marker genes (including the hypersensitive response (HR) marker gene *NtHSR201*, SA-related gene *NtPR3*, ethylene synthesis (ET)-dependent gene *NtAccdeaminase*, JA-related genes *NtLOX1* and *NtDEF1* in the *N. benthamiana* leaves, post-*R. solanacearum* and *F. solani* var. *coeruleum* inoculation for 1 d and 6 d. We used *NtEF-1α* as an internal reference gene. All data points were means ± standard deviation (*n* = 3). Bars that are superscripted by different lower-case letters indicate significant differences, as determined by Duncan’s new multiple range test (*p*-value < 0.05).

**Figure 12 ijms-23-12750-f012:**
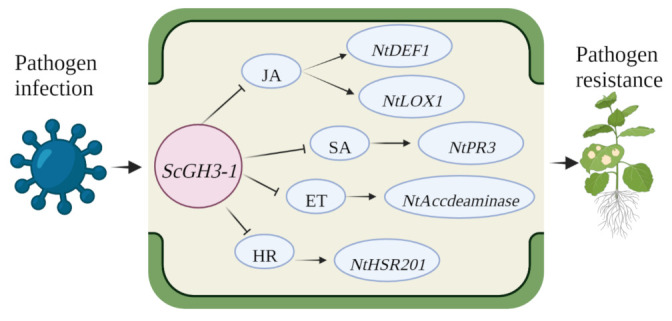
A proposed model for *ScGH3-1*-mediated regulation in the response of *N. benthamiana* to pathogen infection. ET, ethylene synthesis; JA, jasmonic acid; SA, salicylic acid; *NtLOX1*, lipoxygenase 1; *NtDEF1*, defenseless 1; *NtPR3*, pathogenesis-related genes; *NtAccdeaminase*, ACC deaminase gene; *NtHSR201*, hypersensitivity-related genes.

## Data Availability

Not applicable.

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
