# Peer review of "Genome-Wide Identification of Auxin-Responsive GH3 Gene Family in Saccharum and the Expression of ScGH3-1 in Stress Response"

_ijms, 2022, doi:10.3390/ijms232112750_

Round 1
Reviewer 1 Report
Authors have done great work to contribute in scientific research for the improvement of sugarcane resistance against Sporisorium scitamineum stress. They have used smut-resistant and susceptible cultivars YT96-86 and ROC22 respectively. Authors have identified 41 genes belong to ssgh3 gene family and classified them into ssgh3 I, II, and III on the chromosome 19, chr3, and chr7. They have revealed that among 41 genes, 19 pairs of fragmentary duplicated genes in S. spontaneum. Gene cloning demonstrated that sSGH 3 I was constitutively expressed, but when exposed to S. scitamineum inoculation then it was upregulated in susceptible variety ROC22 and downregulated in resistant variety. Overexpression of ssgh3-1 in Benthamiana showed susceptibility to F. solanacearum var. coeruleum and R. solanacearum. Overall, this study provides the basis for improvement of sugarcane crop under disease resistance.
I don’t have many comments, but there are few concerns that should be resolved. For example, in each section of MS, references are not incorporated, refff were missing. It was difficult to understand whether the statement is correct or wrong without checking reference….
I don’t know why you don’t put line numbering? Line numbers always help to comment and point out something.
Methods: Although authors have tried to describe methods section very well, but I think each heading of Methods should be explained for the readers. It’s hard-to-understand If someone wants to follow your methods.
Author Response
Response to Reviewer 1 Comments
Dear reviewer,
We are glad to receive your valuable comments and suggestions and would like to thank you for your kind consideration of our manuscript. The manuscript was revised to address all the opinions, suggestions, and comments and the red fonts were used for revised text. Our point-to-point responses are listed below:
Comment 1: In each section of MS, references are not incorporated, references were missing. It was difficult to understand whether the statement is correct or wrong without checking references.
Author Response: Thanks for your comment. We have revised and added necessary references in the “M&M” section.
Comment 2: I don’t know why you don’t put line numbering? Line numbers always help to comment and point out something.
Author Response: Thanks for your suggestion. We have added the line numbers in the whole manuscript.
Comment 3: Methods: Although authors have tried to describe methods section very well, but I think each heading of Methods should be explained for the readers. It’s hard to understand if someone wants to follow your methods.
Author Response: Thanks for your comment. We have revised the head title of the “Results” and “M&M” sections to be explainary.
We also want to take this opportunity to express our sincere appreciations for your professional review. Your excellent comments will no doubt help improve the quality of our manuscript and guide us in future research endeavor and new manuscript preparation.
Qibin Wu and Youxiong Que
2022-10-12

Reviewer 2 Report
The authors of this article scanned the sugarcane genome and identified 41 SsGH3 genes which were further classified into three groups. They further performed a series of bioinformatics and molecular analysis to establish the importance of some of the Gh3 genes and set up the basis for further study of ScGH3 genes and also pinpoint some candidate genes that can be useful in the molecular breeding of sugarcane for disease resistance.
The manuscript is well written with apt materials and methods and results are discussed well.
The only thing that was tough to asses was the relevance of cited articles because of errors associated with the cited articles "Error! Reference source not found.," I believe it is due to the citation manager used by the authors. Please rectify that.
Author Response
Response to Reviewer 2 Comments
Dear reviewer,
We are glad to receive your valuable comments and suggestions and would like to thank you for your kind consideration of our manuscript. The manuscript was revised to address all the opinions, suggestions, and comments and the red fonts were used for revised text. Our point-to-point responses are listed below:
Comment: The manuscript is well written with apt materials and methods and results are discussed well. The only thing that was tough to asses was the relevance of cited articles because of errors associated with the cited articles "Error! Reference source not found.," I believe it is due to the citation manager used by the authors. Please rectify that.
Author Response: Thanks for your suggestion. We have revised and confirmed that all those references in the revised manuscript are cited correctly.
We also want to take this opportunity to express our sincere appreciations for your professional review. Your excellent comments will no doubt help improve the quality of our manuscript and guide us in future research endeavor and new manuscript preparation.
Qibin Wu and Youxiong Que
2022-10-12

Round 2
Reviewer 1 Report
Dear Authors,
Thank you for following my suggestions and comments. could you please write detail in figure caption of Figure 4? It should stand alone...
Author Response
Dear reviewer, Thanks for your comment. We have added the caption for Figure 4 as below: Figure 4. GO annotation of SsGH3 family genes. GO terms are summarized in three main categories. (A) All GO terms in three main categories. (B, C and D) The functional annotation analysis of SsGH3 genes in biological process, cellular component and molecular function, respectively. Best regards, Qibin Wu and Youxiong Que